# Comparing the Influence of Green Credit on Commercial Bank Profitability in China and Abroad: Empirical Test Based on a Dynamic Panel System Using GMM

**Xiaoling Song [1], Xin Deng [2,*] and Ruixue Wu [3]**

[1] Business School, Beijing Language and Culture University, Beijing 100083, China; sxlhfr@sina.com
[2] School of Finance, Hunan University of Technology and Business; Post-Doctoral Research Station of Management Science and Engineering, National University of Defense Technology, Changsha 410205, China
[3] Beijing Jingdong Shangke Information Technology Co., LTD., Beijing 100086, China; wuruixue2014@163.com
[*] Correspondence: dengxin@hnuc.edu.cn or sunnydoe@163.com

**Abstract:** This study establishes a dynamic panel model for 12 Chinese-listed commercial banks and seven international commercial banks. More specifically, it examines the impact of green credit on the profitability of commercial banks and the differences between China and other countries while using the generalized method of moments. The research shows that the Equatorial Principles project-financing ratio of international banks positively affects bank profitability, while the ratio of green credit for Chinese commercial banks is inversely related to their profitability. Further, a comparative study of China and other countries highlights that the green credit business is at significantly different stages in China and the rest of the world. This study also finds that the profitability of China's banking sector is positively affected by asset size, management expense ratio, cash ratio, and GDP growth rate, in addition to the common influencing factor of non-performing loan ratio, whereas asset size and capital adequacy ratio negatively affects the international banking sector. Drawing on these empirical conclusions, this study offers suggestions for the further development of green credit in Chinese commercial banks.

**Keywords:** green credit; Equator Principles; profitability; dynamic panel data model; GMM

**JEL Classification:** E00; F10; G21

## 1. Introduction

With the worsening ecological environment, the pressure on financial institutions to act in a more socially responsible manner has exponentially increased. At the same time, governments are also aware of the urgency and importance of economic regulation and control for environmental protection. Since 2002, large-scale international banks, such as Citigroup, Barclays, and ABN AMRO, have voluntarily subscribed to the Equator Principles, which provide a framework to assist in resolving environmental issues and project financing irregularities by setting financial standards for environmental risk assessment, measurement, tracking, and loan management.

In general, the key differences between equatorial and non-equatorial banks are in ethics, environmental policies, and bank size (Scholtens and Dam 2007; Zhang 2006). According to the United Nations Environment Programme Finance Initiative, green finance can be divided into four service sectors: retail banking, investment banking, asset management, and insurance. Among these, retail banking provides a large number of financial products, mainly in the form of loans, which can be

further divided into housing mortgage loans, commercial construction loans, auto and transportation loans, and credit card loans (North American Task Force 2007). By March 2017, a total of 89 financial institutions from 37 countries had announced their adoption of the Equator Principles. 43% of these institutions are from Europe, 16% from North America, 10% from Asia (mainly Japan), and 11% from Latin America[1]. Furthermore, two Chinese banks, Industrial Bank and Bank of Jiangsu, announced their adoption of the Principles in 2008 and 2017, respectively.

In 2007, Chinese authorities released a set of recommended policies entitled "Opinions on the Implementation of Environmental Protection Policies and Regulations for Credit Risk Prevention" to ensure the implementation of green credit policies. This was the first time that the implementation and scope of green loans were formally proposed, and the responsibilities and information communication mechanism of departments were formalized in China. Over the last decade, China has also introduced green credit policies, green credit guidelines, and related strategic policies, which have established a preliminary framework for regulating the demand for energy conservation and emissions reduction through financial leverage. By the end of June 2016, the outstanding green credit by the 21 major banking financial institutions in China reached 7.26 trillion Yuan, accounting for 9% of total outstanding loans. Loans for energy conservation and environmental protection, new energy, new energy vehicles, and other strategic emerging industries totaled 1.69 trillion Yuan, and those for energy conservation and environmental protection projects and services amounted to 5.57 trillion Yuan (Institute of Environmental Planning 2017).

Indeed, China has successively introduced strategies and policies that are related to green credit in the past decade, which aim to build a preliminary framework of using financial leverage to regulate the demand for energy conservation and emission reduction in the environment industry. However, there are many differences between China's framework and the international norms regarding the rules as compared to the international norms of corporate social responsibility (CSR) and industry standards of green credit. Research on the impact of the development of green credit business on commercial banks in China has not yet come to a clear consensus due to these complexities. This paper established two panel data models that are treated differently, rather than mixing them together to address this challenge. Subsequently, the following four questions naturally arise: (1) what is the mechanism through which green credit influences the profitability of commercial banks under the dual objectives of public welfare and profitability? (2) does the green credit business of commercial banks affect their profitability? (3) is this effect significant? and (4) what are the differences between Chinese and international commercial banks? This study addresses these questions in detail.

This research contributes to the literature regarding the correlations between green credit and banking performance as well as to the literature on comparisons between policy implementation of green credits in China and corporate social responsibility (CSR) practices worldwide. Moreover, the results of our comparative analysis allow for us to find the gap between China's green credit business and that at the international level, so as to find an effective way to better balance the Chinese banks' profit objectives and corporate social responsibilities.

The remainder of the paper is organized, as follows. Section 2 provides a literature review. Section 3 conducts a theoretical analysis of the impact of green credit on the profitability of commercial banks. Section 4 performs an empirical analysis and, accordingly, constructs dynamic panel models for the sample of Chinese and international banks. The purpose is to analyze the impact of green credit and related indicators on commercial bank profitability while using the generalized method of moment (GMM). Section 5 offers conclusions and suggestions regarding the future development of green credit in China.

---

1　See the Equator Principles website: http://www.equator-principles.com/index.php/ep3.

## 2. Literature Review

Studies have long discussed the impact of the financial sector on the environment. Richardson (2005) suggests that enterprises in OECD countries have relied on external financing since the 1970s. To a certain extent, financiers are responsible for the environmental damage that is caused by polluting enterprises. The financial sector does not directly influence the environment, but it has an indirect effect by providing investment loans to enterprises that cause environmental damage. Allen and Yago (2011) indicate that to build a more sustainable economy, financial technologies should be applied to evaluate and price environmental externalities (e.g., water, air pollution, and public resources). Ning and She (2014) studied the relationship between green finance and the level of macroeconomic development in China from 1978 to 2010 and found that, while there is a long-term stable equilibrium relationship between them, the development scale for green finance and the efficiency of resource allocation have an inhibitory effect on macroeconomic development. Some studies have also examined the correlation between the CSR development index and the business performance of banks and other financial institutions in China (Yin et al. 2014) and other countries (Simpson and Kohers 2002).

With a sharp increase in implementation of CSR practices worldwide (Engle 2007; Vartiak 2016; McPherson 2018), researchers have turned their attention to the relationship between CSR and the financial performance of financial intermediates. Numerous studies suggest that the application of CSR, such as the adoption of Equator Principles or the development of green credit, can improve banks' profitability. Eisenbach et al. (2014) study the performance of large-scale financial institutions of various countries in the global project financing market and show that financial institutions that adopt the Equator Principles have positive excess returns in terms of market share and other aspects. Li and Wang (2014) analyze 10 equatorial banks and find that the amount of financing by financial institutions in low-risk (or level C) projects, infrastructure, renewable energy, and emerging markets is positively correlated with profitability. Malik et al. (2015) also found a positive relationship between an awareness of CSR activities and Pakistan's organizational performance. Brogi and Lagasio (2019) found a significant and positive association between environmental, social, and governance activities, as well as evidence that the environmental awareness in United States (US) banks is strongly related to profitability, providing implications for policy makers and policy takers. A study by Maqbool and Zameer (2018) provided an empirical analysis of Indian banks' CSR activities and their financial performance. Their results provide great insights for management regarding the integration of CSR with the strategic intent of a business and the renovation of business philosophies from a traditional profit-oriented approach to a socially responsible approach.

In recent years, with numerous banks disclosing green credit activities in China, more studies have turned their focus on the relationship between corporate sustainability and financial indicators of Chinese commercial banks. Numerous studies have found that green credit has positive effects on banking operations. In line with institutional theory, the Chinese Green Credit Guidelines might influence both corporate sustainability performance and financial performance of Chinese banks by exposing them to coercive pressure (Phan and Baird 2015). Weber (2017) examined reports and websites of Chinese banks, categorized different corporate sustainability features, and conducted panel regression and Granger causality to analyze cause and effect variables. The environmental and social performance of Chinese banks significantly increased between 2009 and 2013. Furthermore, a bi-directional causality between financial performance and sustainability performance of Chinese banks has also been found. Sun et al. (2017) and Liao et al. (2019) empirically found that when controlling for variables, such as the non-performing loan ratio and total assets, green credit positively affects the operational efficiency of banks in the short term, and this effect stabilizes in the long term. Cui et al. (2018) used panel regression techniques, including two-stage least square regression analysis and random-effect panel regression, to examine whether a higher green credit ratio reduces a bank's non-performing loan ratio based on a five-year dataset of 24 Chinese banks (NPL ratio). The results suggested that allocating more green loans to the total loan portfolio does reduce a bank's NPL ratio and

that institutional pressure by the Chinese Green Credit Policy has a positive effect on the environmental and financial performance of banks.

Some other studies have found that green credit has an uncertain or even negative effect on bank profitability. Researchers, including Scholtens and Dam (2007) and Wright (2012), as well as Chinese scholars, such as Hu and Zhang (2016) and Han et al. (2017), conclude that green credit has special traits, including prolonged periods of project construction, significant investment amounts, and high risk of policy change. Therefore, specialized green credit only increases operating costs, which results in a decline in operating efficiency, especially in the short term. From an empirical perspective, Wang and Zhu (2017) examined China's listed banks and demonstrate an inverted U-shaped relationship between green credit and bank performance. When the proportion of green credit is low, the substitution effect is significant and bank performance improves. On the other hand, when the proportion is high, the crowding-out and risk amplification effects gradually appear and bank performance declines. Chen and Dong (2008) state that the withdrawal of loans from highly polluting industries, or those with excess capacity in recent years, detrimentally affects the profitability of Chinese banks.

In summary, there is a general consensus in the literature that, with the development of a green credit business scale, the influence of green credit on profitability, and even the competitiveness of the banking industry will become increasingly significant in the future. However, findings on the influence of adopting the Equator Principles or of promoting green credit business on commercial bank profitability in the short term remain inconsistent. While existing research has made several important contributions, a few aspects warrant further investigation. First, the mechanism of green credit's influence on the profitability of commercial banks must be further analyzed. Second, research on equatorial banks has only examined a limited range of banks, while the Equatorial Principles are subject to loopholes and arbitrary behaviors. Further, the current research appears to be concentrated on banks in a certain country or region. Few studies empirically test global samples for the impact of green credit on commercial banks' profitability and compare cross-country differences.

This study attempts to address the abovementioned gaps in the literature in the following manner. First, in line with several aspects, including product expansion, expected risk, social reputation, and comprehensive competitiveness, it elucidates the impact mechanism for the development of green credit businesses on commercial banks' profitability and, accordingly, proposes a number of basic theoretical assumptions. Second, we select are presentative sample of commercial banks from China and other countries and establish a two-panel data model, adopting GMM to study the similarities and differences between the key factors that affect commercial bank profitability in China and abroad. We then focus on the direction and extent of green credit's influence. Finally, from the empirical conclusions, this study seeks a new explanation for the moderate expansion of green credit in China and offers suggestions for the development of green credit in the country.

In sum, the study contributes to the current knowledge on the association between banking performance and green credit in addition to the available literature that compares the policy implementation of green credits in China and CSR practices worldwide. Moreover, our comparative analysis illustrates the gap between China's green credit business and that, at the international level; this can help policy makers to find effective ways to better balance the Chinese banks' profit objectives and corporate social responsibilities.

## 3. Theoretical Analysis and Basic Assumptions

Green credit originated in the field of green finance, which seeks to understand the economic means banks use to face resource depletion. Through green credit, banks take into consideration the environmental friendliness and law-abiding status of enterprises in the accounting and approval of financial loans. More specifically, green credit enforces strict control over loans that are provided to enterprises producing excessive pollution levels and provides preferential policy loans to those with higher efficiency and cleaner production methods. By limiting the amount of money available to high-polluting projects and using financial instruments with punitively high interest rates, green

credit routes funds toward environmentally friendly enterprises, helping to achieve the objective of a more environmentally friendly economy. Theoretically, the influence of green credit on commercial bank profitability extends, in turn, to product expansion, expected risks, social reputation, and comprehensive competitiveness (see Figure 1).

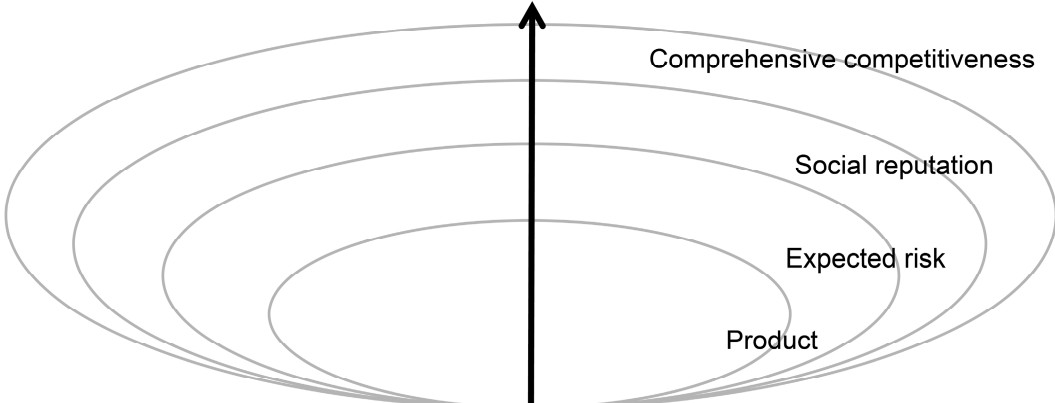

**Figure 1.** Influence transmission mechanism for green credit and commercial bank profitability.

### 3.1. Role of Green Credit in Banking Services and New Revenue Growth Points

First, a green credit business chain will be largely involved in supporting businesses, cover project management, qualification verification, R&D, fund allocation, and post-loan management process. Consequently, the growth of green credit can attract relevant intermediary businesses (e.g., settlement and delivery, guarantee, custody, and consulting) and, therefore, serves as a new profit source for the middle businesses of commercial banks. Second, green credit can affect the original structure of bank loans and create a more diversified loan business. For example, Fannie Mac and Citigroup in the United States introduced energy-saving housing mortgage loans, CFS in the United Kingdom launched ecological home loans, Rabobank in the Netherlands issued a climate credit card, and China's Industrial Bank launched low-carbon credit cards. Such innovation in financial services can generate more banking profits (North American Task Force 2007). At the same time, commercial banks can increase the liquidity of green credits through green asset mortgages, so that previously unliquidated green assets may become productive again, thus improving bank profitability. By early 2016, Industrial Bank issued China's first green credit asset-backed securities for a total amount of 2.6457 billion Chinese Yuan. The asset pool contained 42 green finance loans that effectively revitalized the bank's loan funds.[2]

**Hypothesis 1.** *The growth of green credit businesses can increase bank income and, consequently, improve profitability.*

### 3.2. Effect of Green Credit on Commercial Bank Expected Credit Risk Level

Loan projects for green credit mainly include energy conservation and environmental protection projects. Commercial banks can reduce their non-market credit risk if they adopt such projects as part of their development policy. Prior to the issuance of green loans, commercial banks perform comprehensive due diligence, evaluating borrowers to assess environmental and social risks, followed by a dynamic analysis. Thus, strict credit management can reduce the default risk that is caused by environmental problems and maintain the non-performing loan ratio at a low level, improving the business outlook of commercial banks (Wörsdörfer 2015). Green credit projects are less risky when

---

[2]    See the 2016 annual report by China's Industrial Bank.

compared with traditional projects, and they can effectively curb the rise of non-performing loan ratios and increase expected earnings. Therefore, the development of green credit by commercial banks should help them to enhance their ability to withstand risks, thus improving their net profit and non-interest income (Industrial and Commercial Bank of China Environmental Factor Stress Test Research Group 2016; Sun et al. 2017).

**Hypothesis 2.** *The growth of green credit businesses reduces the expected credit risk of banks, thus improving bank profitability.*

*3.3. Green Credit Enhances Commercial Banks' Responsible Social Image and Market Competitiveness*

The development of green credit positively affects both environmental protection and financial industry development. First, commercial banks are obligated to assume corporate social responsibility, that is, while meeting profit demands, they must preserve the benign development of social environment, including rationally utilizing resources, reducing environmental damage, and curbing energy consumption and pollution emissions. When compared to their market peers, commercial banks that offer green credit will gain more business resources because of their moral and environmental stance, which could not only win them a large market share, but also promote the virtuous circle of capital, consequently improving their profitability (Cowton and Thompson 2000). Second, by referencing the Equator Principles, banks can learn to balance their social responsibility and economic targets. Bank shareholders are unlikely to negatively react to the adoption of the Equator Principles; rather, they consider it to be an indicator of a bank's social responsibility (Scholtens and Dam 2007). Therefore, the combination of bank profitability and social responsibility can improve social reputation and create new profit growth points for banks, enhancing their market competitiveness (Eshet 2017; Wang 2016).

**Hypothesis 3.** *The growth of green credit businesses improves bank social reputation, and therefore, enhances their profitability.*

According to marginal cost and revenue analysis theory, a commercial bank's marginal cost (MC) and marginal revenue (MR) curves generally intersect at point A, at which the bank's profit is maximized, and P and Q are the equilibrium price and quantity, respectively. Growth in the size of green credit generates positive externalities on related banking products, credit risk, social image, and comprehensive competitiveness. In this case, the marginal revenue curve will move up to MR' and intersect MC at A', causing the equilibrium price and quantity to rise to P' and Q', respectively. In sum, the overall profitability of the bank increases, as shown in Figure 2.

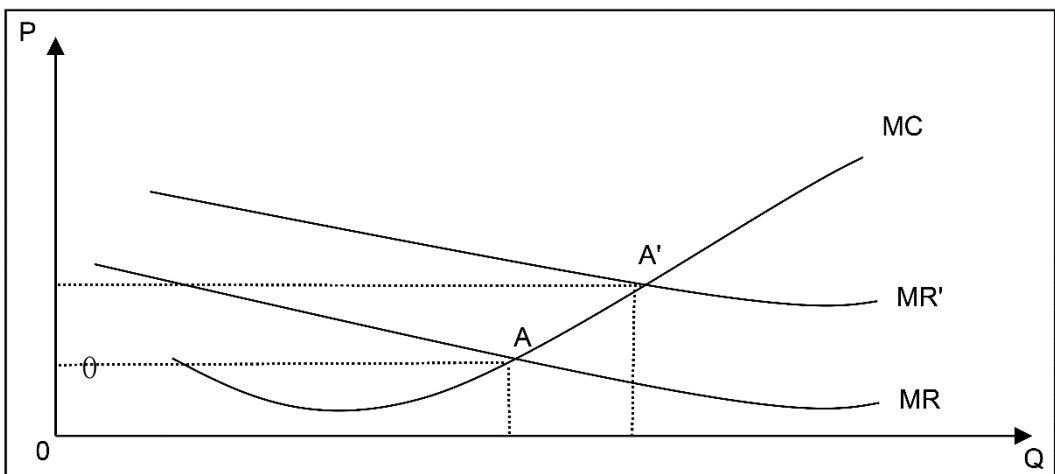

**Figure 2.** Marginal cost and revenue analysis of commercial bank loans when green credit increases.

## 4. Empirical Analysis

As mentioned in the introduction, we use two dynamic panel data models since the development level and green credit business practices of China's commercial banks are inconsistent with those of international banks. In addition, we compare the impact of green credit on commercial banks profitability in China and other countries by examining the major endogenous and exogenous factors affecting commercial bank profitability.

### 4.1. Data Sources and Variable Selection

This study examines data from the first quarter of 2008 to the fourth quarter of 2015, selecting the following 12 banks that publicly disclose green credit information: Industrial and Commercial Bank of China, Agricultural Bank of China, Bank of China, China Construction Bank, Bank of Communications, China Everbright Bank, Huaxia Bank, Ping An Bank, China Industrial Bank, eChina Merchants Bank, Shanghai Pudong Development Bank, and China CITIC Bank. For comparison, we selected seven equator banks in other countries that publicly disclose loan sizes in line with the Equator Principles: HSBC, Standard Chartered Bank, Citibank, Barclays, Banco Bilbao Vizcaya Argentaria (Spain), Bradesco Bank (Brazil), and SAS bank (Sweden). The data in this study are taken from each bank's annual and social responsibility reports that were published on their official websites. GDP data are obtained from the World Bank database. We used EViews 7.2 as statistical analysis software.

There are some limitations of the data sample based on the data availability. First, the green credit business in China developed relatively late as compared to that in Western countries; China's banking industry did not begin to strengthen the disclosure of green credit data until 2008. Thus, this study examines data from the first quarter of 2008 to the fourth quarter of 2015. Second, there are only 22 listed commercial banks in China, and we were only able to select 12 banks that publicly disclose green credit information. Additionally, we selected seven equator banks in other countries that publicly disclose loan sizes that are in line with the Equator Principles. According to the equator principles, most international banks only disclose the aggregate project amount for three risk levels: level A (high risk), level B (medium risk), and level C (low risk), rather than a specific amount for equator loans, so we can seldom collect the data of equator loan amount. Methodologically, this study draws from works by Yang (2008) and Wang (2011). In addition to focusing on green credit, we have introduced several control variables, such as non-performing loan ratio and capital adequacy ratio. The following subsection describes these variables in detail.

### 4.1.1. Explained Variable

Numerous indicators reflect the profitability of commercial banks, such as return on total assets (ROA), return on equity (ROE), and operating profit margin. Commercial banks earn most their income from interest rate spreads between deposits and loans, and highly leveraged business models. Among other factors, such as foreign exchange service, intermediate service, and security investment, this study uses ROA as the explained variable to measure bank profitability.

### 4.1.2. Explanatory Variables

The green credit loan ratio (GCLR), also known as the project-financing ratio under the Equator Principles, is calculated as the green credit balance (Equator Principles project financing) divided by the total loan amount. The literature offers two primary measures for green credit: the first is the total amount of green credit balance e.g., (Wang and Zhu 2017) and second is a relative indicator for the green credit ratio. We select a relative indicator to improve the comparability of the results as total asset size in our sample is not comparable across countries due to differences in banking systems and other factors.

### 4.1.3. Control Variables

This study controls for the following endogenous factors affecting banking profitability:

First, NPL is estimated as the ratio of non-performing loans to total loans. A high ratio indicates that the bank's asset quality is low and negatively affects the bank's sustainable profitability.

Second, the capital adequacy ratio (CAR) is calculated as the ratio of total capital to risk-weighted assets. Under the 2010 Basel III rules, CAR should be maintained at 8%, but the Tier 1 standard has been raised to 6%. The capital adequacy ratio represents a bank's solvency against risks by helping to set a stable profit level. However, an excessively high CAR indicates that banks do not fully utilize capital for business operations and expansion activities, so their profitability will gradually weaken in the future.

Third, management fees (MEP) are calculated as the ratio of administrative expenses to operating expenses. Management costs, such as staff salaries, daily office expenses, equipment renewal fees, and other fixed expenses, measure management efficiency. Against the background of fierce competition in the banking sector, commercial banks must constantly improve their management efficiency and profitability. High MEP indicates low management efficiency as relatively high management costs negatively impact profitability.

Fourth, the cash ratio (CR) is the ratio of cash to total assets. Asset liquidity refers to a bank's ability to liquidate assets at a reasonable price to meet short-term customer cash demand, generally using cash assets and short-term bills or loans. A high CR denotes quick short-term solvency, although excessive liquidity dampens the profitability of long-term assets.

Finally, we use total assets (STA) after standardization. The impact of asset size on commercial bank profitability is reflected in the effect of economies of scale on operating costs. With an increase in asset size, the average banking cost follows a U-shaped distribution, thus affecting economic benefits. In sum, greater total assets improve the bank's operations and increase its market share and resources.

This study also controls for GDP growth rate as an exogenous variable that affects bank profitability. The level of macroeconomic development has substantial impact on commercial bank profitability. Under favorable economic conditions, enterprises are willing to expand their businesses and, consequently, there is relatively greater loan demand. This situation brings abundant high-interest loan projects to banks and increases enterprise loan income. In addition, policies, regulations, and other external factors, such as institutional factors and historical path, affect the business standards of commercial banks and, thus, their profitability. However, it is difficult to quantify policy impact, because policies and regulations differ by country, and banks across countries can adapt their operations to the regulatory environment. Thus, this study does not include policy factors in the empirical analysis. Table 1 shows the descriptive statistics for all of the variables.

**Table 1.** Descriptive statistics.

| Variable Symbol | Variable | Sample Size | Mean | Variance | Minimum | Maximum |
|---|---|---|---|---|---|---|
| | | Chinese commercial bank (Model 1) | | | | |
| ROA | Return on total assets | 96 | 0.010 | 0.002 | 0.001 | 0.005 |
| GCLR | Green credit loan ratio (Equator Principles project financing ratio) | 82 | 0.080 | 0.019 | 0.010 | 0.071 |
| NPL | Non-performing loans | 96 | 0.012 | 0.006 | 0.004 | −0.008 |
| CAR | Capital adequacy ratio | 96 | 0.128 | 0.053 | 0.086 | 0.063 |
| MEP | Management fee | 96 | 0.616 | 0.125 | 0.055 | −0.11 |
| CR | Cash ratio | 96 | 0.143 | 0.026 | 0.081 | 0.1 |
| STA | Total assets | 96 | 0.000 | 1.000 | −1.028 | 0 |
| GDP | GDP growth rate | 96 | 0.086 | 0.013 | 0.069 | 0.079 |
| | | International commercial bank (Model 2) | | | | |
| ROA | Return on total assets | 56 | 0.005 | 0.004 | −0.009 | 0.017 |
| GCLR | Green credit loan ratio (Equator Principles project financing ratio) | 54 | 0.009 | 0.007 | 0.001 | 0.041 |

| Variable Symbol | Variable | Sample Size | Mean | Variance | Minimum | Maximum |
|---|---|---|---|---|---|---|
| | | International commercial bank (Model 2) | | | | |
| NPL | Non-performing loans | 56 | 0.020 | 0.026 | 0.001 | 0.121 |
| CAR | Capital adequacy ratio | 56 | 0.065 | 0.021 | 0.018 | 0.128 |
| MEP | Management fee | 56 | 0.726 | 0.230 | 0.316 | 0.945 |
| CR | Cash ratio | 56 | 0.043 | 0.027 | 0.012 | 0.134 |
| STA | Total assets | 56 | 0.000 | 1.000 | −1.294 | 1.532 |
| GDP | GDP growth rate | 56 | 0.007 | 0.022 | −0.043 | 0.075 |

## 4.2. Stationarity Tests

A regression model used on non-stationary time series data can easily result in a spurious regression. We perform stationary tests to avoid this issue. There are five commonly used test methods: Levin–Lin–Chu (LLC); IM, Pesaran, and Shin's (IPS) W-stat; Breitung; ADF-Fisher; and, PP-Fisher. These methods assume that the variable contains a unit root and, therefore, is non-stationary. If the null hypothesis is rejected, then the data is stationary.

We consider three models in this study, namely, models with intercept, models with intercept and trend, and models with no intercept or trend. We begin with the intercept and trend model, and then consider the intercept model, followed by the model with no trend and intercept. Models are denoted by triplets of the form $(a, b, c)$, where $a$ might be i or 0, depending on whether there is an intercept term for bank i, $b$ may be t or 0, depending on whether there is a time trend for bank i, and c is always zero since no moving average component is assumed. If a model rejects the assumption of a unit root, the sequence is considered to be stationary.

$$\text{Intercept and trend model } (i, 0, 0): \Delta y_{it} = \alpha_{0i} + \delta \Delta y_{it-1} + \varepsilon_{it}. \tag{1}$$

$$\text{Intercept model } (i, t, 0): \Delta y_{it} = \alpha_{0i} + \alpha_{1i} t + \delta \Delta y_{it-1} + \varepsilon_{it}. \tag{2}$$

$$\text{Model with no intercept or trend } (0, 0, 0): \Delta y_{it} = \delta \Delta y_{it-1} + \varepsilon_{it}. \tag{3}$$

Table 2 presents the test results. The data for Chinese and international commercial banks are both stationary time series.

**Table 2.** Stationarity test results.

| Model | Variable | Test(s) Passed | Test Results | Data Model |
|---|---|---|---|---|
| Chinese Commercial Banks | ROA1 | LLC, IPS, ADF, PP | stationary | (i, 0, 0) |
| | GCLR1 | LLC, PP | stationary | (i, t, 0) |
| | NPL1 | LLC, PP | stationary | (i, t, 0) |
| | CAR1 | LLC, PP | stationary | (i, t, 0) |
| | MEP1 | LLC, PP | stationary | (i, t, 0) |
| | CR1 | LLC | stationary | (0,0,0) |
| | STA1 | LLC, PP | stationary | (i, t, 0) |
| | GDP1 | LLC, Breitung, IPS, ADF | stationary | (i, t, 0) |
| International commercial banks | ROA2 | LLC, IPS, ADF, PP | stationary | (i, t, 0) |
| | GCLR2 | LLC, PP | stationary | (i, t, 0) |
| | NPL2 | LLC, IPS, ADF, PP | stationary | (i, t, 0) |
| | CAR2 | PP | stationary | (i, 0, 0) |
| | MEP2 | LLC, PP | stationary | (i, 0, 0) |
| | CR2 | LLC, PP | stationary | (i, t, 0) |
| | STA2 | LLC | stationary | (i, t, 0) |
| | GDP2 | LLC, ADF, PP | stationary | (i, t, 0) |
| | LNCAR2 | LLC, PP | stationary | (0, 0, 0) |

*4.3. Model Building*

According to the theoretical analysis presented thus far, green credit affects commercial bank profitability through multiple channels, mostly by using time serials data or panel data. However, economic theory tells us that bank profit is a continuous dynamic process, so the profitability from the previous period would have some impact on the current period; thus, we introduced a lag dependent variable that more accurately conforms to the theory and reality. However, once the lag dependent variable is introduced into the equation, the original static model is transformed into a dynamic model. It is only by using the dynamic panel data model that the accuracy of the conclusion can be guaranteed. Therefore, this paper established two models: one for the Chinese banking industry and one for banks from other countries. We adopted a dynamic panel data model based on generalized moment estimation by introducing the lag term of ROA to more effectively measure the influence of green credit.

This study aims to directly investigate the relationship between ROA and green credit ratio by constructing dynamic panel data models.

At the same time, when considering that ROA is affected not only by various factors in the same period, but also its past value, we introduce a phase-1 lag term for ROA in the model and control variables that impact bank profitability. Thus, we use the following dynamic panel model:

$$ROA_{it} = \alpha_1 ROA_{it-1} + \sum_{i=2}^{n} \alpha_i X_{kit} + \mu_i + \varepsilon_{it}, \tag{4}$$

where $ROA_{i_t}$ is the dependent variable, $ROA_{i_{t-1}}$ is the dependent variable lagged in phase 1, $X_{ki_t}$ is the independent variable, $\alpha_i$ is the independent variable coefficient, $\mu_i$ denotes individual effects, and $\varepsilon_{it}$ is a random error term. We expect a correlation between the phase-1 lagged term of the dependent variable and the individual effect, which will cause endogenous problems. This study adopts the generalized method of moments (GMM) for the estimation since the least square method cannot be used for a regression in this model.

GMM can be used to estimate population and sample moments. We use the first-order origin moment, $X^1$, of the samples as an estimator for the first-order population moment ($M^1$) along with the second-order central moment ($X^2$) as an estimator for the second-order population moment ($M^2$). The estimators for the first and second moments are as follows:

$$\text{First moment of population}: \quad \hat{M}^1 = \hat{X}^1 = \frac{1}{n} \sum_{i=1}^{n} ROA_i. \tag{5}$$

$$\text{Second moment of population}: \quad \hat{M}^2 = \hat{X}^2 = \frac{1}{n} \sum_{i=1}^{n} ROA_i{}^2. \tag{6}$$

The expectations and variances of the population parameters are estimated as follows:

$$\text{Expectation of population parameters}: \quad \hat{\mu} = \hat{M}^1 = \frac{1}{n} \sum_{i=1}^{n} ROA_i = \overline{ROA} \, q. \tag{7}$$

Variance of population parameters:

$$\hat{\sigma}^2 = \hat{M}^2 - \left(\hat{M}^1\right)^2 = \hat{X}^2 - \left(\hat{X}^1\right)^2 = \frac{1}{n} \sum_{i=1}^{n} ROA_i{}^2 - \left(\frac{1}{n} \sum_{i=1}^{n} ROA_i\right)^2$$
$$= \frac{1}{n} \sum_{i=1}^{n} \left(ROA_i - \overline{ROA}\right)^2. \tag{8}$$

GMM should be used when the number of moment estimation equations that are selected is larger than the parameters to be estimated:

$$
\begin{aligned}
X^1 - M^1\big(\beta_1,\ \beta_2,\ \cdots \beta_r\big) &= 0 \\
X^2 - M^2\big(\beta_1,\ \beta_2,\ \cdots \beta_r\big) &= 0 \\
&\vdots \\
X^r - M^r\big(\beta_1,\ \beta_2,\ \cdots \beta_r\big) &= 0
\end{aligned}
\tag{9}
$$

To evaluate the parameter $\beta$, it is necessary to minimize the Euclidean distance function:

$$
Q\big(\hat{\beta}\big) = \sum_{i=1}^{r}\big(X^i - M^i(\beta)\big)^2.
\tag{10}
$$

The weighted least squares method can enhance the effect of specific moments in the moment estimation. Denoting the sample and population while using vectors: $X = \big(X^i,\dots,X^r\big)^T$, $M = \big(M^i,\dots,M^r\big)^T$, the beta for the GMM parameter estimation must minimize the following forms of $Q\,(\beta)$, where $S$ is the covariance matrix for $(X - M)$:

$$
Q\big(\hat{\beta}\big) = (X - M)^T S^{-1}(X - M).
\tag{11}
$$

When the explanatory variable in a model is dependent on the random error, the solution is to replace the random explanatory variable with an instrumental variable Z, which must be independent of $\varepsilon$. Hence, in the model $Y_i = f(X_i, b) + \varepsilon_i$, $b$ is the parameter vector to be estimated, as follows:

$$
\begin{cases}
\varepsilon(Y_i, X_i; b) = Y_i - f(X_i; b) \\
m(b) = \frac{1}{n}\sum_i Z_i \varepsilon(Y_i, X_i; b) = Z^T \varepsilon(Y_i, X_i; b)
\end{cases}.
\tag{12}
$$

$(b) = 0$ is the instrumental variable method of estimation for the parameters. $m\,(b)$ can also be written, as follows:

$$
m(b) = \begin{pmatrix} m_1(b) \\ m_2(b) \\ \dots \\ m_k(b) \end{pmatrix} = \begin{pmatrix} \frac{1}{n}\sum_i Z_{1i}\varepsilon_i \\ \frac{1}{n}\sum_i Z_{2i}\varepsilon_i \\ \dots \\ \frac{1}{n}\sum_i Z_{ki}\varepsilon_i \end{pmatrix}.
\tag{13}
$$

*4.4. Regression Results and Sargan Test*

First, we regress the data for model 1 while using a stepwise regression method and find that only the coefficient for CAR1 is not significant. Table 3 presents the regression results for the other variables after excluding CAR1:

Similarly, we use a stepwise regression for model 2 and find three non-significant variables: CR2, MEP2, and GDP2. Table 4 presents the regression results, excluding variables.

The Sargan test can be used to examine for excessive identification constraints in a statistical model. The null hypothesis is that the model is excessively constrained correctly. The results for the Chinese model indicate that the *p*-value for the *j*-statistic is 0.503, which is significantly greater than 0.1. This indicates that the hypothesis cannot be rejected at the 10% significance level, that is, the model is correctly set. Therefore, model 1 can be expressed, as follows:

$$
\text{ROA}_1 = 0.513\text{ROA}_{1t-1} - 0.027\text{GCLR}_1 - 0.084\text{NPL}_1 + 0.006\text{MEP}_1 + 0.03\text{CR}_1 + 0.003\text{STA}_1 + 0.04\text{GDP}.
\tag{14}
$$

The results for international commercial banks report that the *p*-value for the *j*-statistic is 0.364 and, thus, greater than 0.1, which indicates that, for model 2 also, the null hypothesis cannot be rejected at the 10% significance level and, thus, model 2 is correctly set. Model 2 is given by:

$$ROA_2 = 0.381ROA_{2t-1} + 0.135GCLR_2 + 0.103NPL_2 - 0.004STA_2 - 0.12CAR_2. \tag{15}$$

The regression results show that the explanatory variable, green credit ratio or Equatorial Principles project financing ratio, in the two models significantly impact bank profitability at the 1% significant level, but the effects run in opposite directions for both models.

**Table 3.** Model 1 regression results for Chinese commercial banks.

| Variables | Coefficient | Standard Error | *t*-Value | *p*-Value |
|---|---|---|---|---|
| ROA1(−1) | 0.513 *** | 0.193 | 2.653 | 0.0100 |
| GCLR1 | −0.027 *** | 0.008 | −3.393 | 0.0012 |
| NPL1 | −0.084 | 0.052 | −1.614 | 0.1113 |
| MEP1 | 0.006 ** | 0.003 | 2.025 | 0.0469 |
| CR1 | 0.030 *** | 0.007 | 4.331 | 0.0001 |
| STA1 | 0.003 *** | 0.000 | 3.152 | 0.0025 |
| GDP1 | 0.040 ** | 0.018 | 2.192 | 0.0320 |
| Mean dependent variable | 0.000253 | S.D. dependent variable | | 0.001022 |
| S.E. of regression | 0.001373 | Sum squared residual | | 0.000123 |
| *j*-statistic | 4.332194 | Instrument rank | | 12 |
| prob (j-statistic) | 0.502644 | | | |

Note: ***, **, and * denote significance at the 1%, 5%, and 10% level, respectively.

**Table 4.** Model 2 regression results for international commercial banks.

| Variables | Coefficient | Standard Error | *t*-Value | *p*-Value |
|---|---|---|---|---|
| ROA2(-1) | 0.381 ** | 0.160 | 2.379 | 0.0226 |
| GCLR2 | 0.135 *** | 0.043 | 3.176 | 0.003 |
| STA2 | −0.004 *** | 0.001 | −3.103 | 0.0037 |
| CAR2 | −0.120 * | 0.069 | −1.754 | 0.0878 |
| NPL2 | 0.103 | 0.067 | 1.527 | 0.1352 |
| Mean dependent variable | −0.000183 | S.D. dependent variable | | 0.003183 |
| S.E. of regression | 0.003287 | Sum squared residual | | 0.0004 |
| *j*-statistic | 2.018491 | Instrument rank | | 7 |
| prob(*j*-statistic) | 0.364494 | | | |

Note: ***, **, and * denote significance at the 1%, 5%, and 10% level, respectively.

Model 1 indicates that, for each unit increase in the green credit ratio for Chinese commercial banks, the ROA decreases by 0.027 units. For model 2, on the other hand, a one-unit increase in the Equatorial Principles project-financing ratio for international commercial banks increases ROA by 0.135 units. Thus, model 2 confirms the three basic assumptions, whereas model 1 contradicts them.

We compare the regression results for the two models and divide the control variables into three categories. The first type of variable significantly affects the Chinese and international banks in the same direction. In both the models, ROA in the previous period is the only factor to have significant influence and positive correlation. For every unit of ROA increase in the previous period, the ROA for Chinese commercial banks in the next period increases by 0.513 units. However, for international banks, the ROA only increases by 0.381 units, which indicates that this variable plays a marginally smaller role. Nevertheless, for both the Chinese and the international banks, an appropriate retained profits policy can supplement a bank's capital and offer the advantage of reasonable asset reallocation. These conditions are conducive to the development of banking businesses and widening the profit space.

The second type of variable influences both Chinese and international bank, but in different directions. This category includes non-performing loan ratio and total assets. For every unit increase in the non-performing loan ratio, the ROA of the Chinese banks declines by 0.084 units, while, for the international bank banks, it increases by 0.103 units. However, the *p*-value for the NPL variable in both models is slightly higher than 10%, indicating a less important role in explaining banks profitability. Next, for every unit increase in total asset size, the ROA of Chinese commercial banks significantly increases by 0.003, while, for the international banks, it decreases by 0.004 units. This is because Chinese bank assets are concentrated in state-owned and joint-stock banks, whose market positions are so prominent that they can widen their profit space. However, the competition among international banks is fierce and the expansion in the asset scale cannot lead to decisive resource orientation. Therefore, the growth of bank scale does not generate higher operating income; instead, it increases the operating pressure.

Finally, the third type of variable is the independent influencing factor. The significantly positive effects of management expense ratio, cash ratio, and GDP growth rate also influence Chinese commercial banks. For every unit of increase in these variables, ROA increases by 0.006, 0.03, and 0.04 units, respectively. In theory, excessive management expenses, such as employee salaries, welfare benefits, and publicity fees, reduce ROA. However, the positive correlation result for the Chinese banks indicates that the management efficiency of Chinese commercial banks is at a reasonable level and an appropriate increase in the management expense ratio would help to improve their operational efficiency and increase their ROA. In addition, a greater cash ratio results in better liquidity and higher short-term solvency, which positively affect future earnings. GDP growth rate represents the macroeconomic status of a country. The steady growth of GDP is beneficial in the development of commercial banks and it can increase the rate of return, irrespective of the nature of investment (i.e., real economy or financial sector), thus helping banks to maintain a high ROA.

International banks are negatively affected by the capital adequacy ratio (10% significance level), as an increase of one unit in this variable decreases ROA by 0.12 units. This shows that a higher capital adequacy ratio for international banks is not conducive to the healthy and sustainable development and it will reduce their profitability.

## 5. Conclusions and Suggestions

This study examined 12 listed Chinese banks that disclose green credit information and seven international banks following the Equator Principles from the first quarter of 2008 to the fourth quarter of 2015. More specifically, we conducted a theoretical analysis of the mechanism through which green credit affects commercial bank profitability. To this end, we used the GMM approach to construct a dynamic panel system for empirically testing the impact of green credit on the profitability of Chinese and international banks. The results show that, first, following the Equator Principles positively impacts bank profitability, which confirms the basic hypothesis that green credit improves bank profitability. Second, there is an inverse relationship between the green credit ratio of Chinese commercial banks and their profitability. This may seem to be inconsistent with the hypothesis, but, in fact, the contrast reflects the different development stages of green credit businesses in China and abroad. Finally, in addition to the common influence that is exerted by the non-performing loan ratio, the profitability of China's banking industry is positively affected by asset size, management expense ratio, cash ratio, and GDP growth rate, whereas international bank profitability is negatively affected by asset size and capital adequacy ratios.

The main limitation of this paper derives from its sample size. In particular, international banks seldom announce a specific amount for their equator loans; instead, they only tend to disclose the aggregate project amount for three risk levels: level A (high risk), level B (medium risk), and level C (low risk). In the future, with a higher degree of data disclosure, subsequent scholars will be able to collect data from more international commercial banks with a similar scale of assets. Additionally,

we would be able to select a more comparable variable regarding green credit or CSR and analyze its effects multi-dimensionally, including the bank's profitability, risk, and liquidity.

We propose the following recommendations drawing on these conclusions.

First, international banks following the Equatorial Principles have a long history of development and enjoy the obvious advantages of project inspection, qualification examination, capital allocation; business operation processes, business innovation, technical personnel training, and risk assessment system. China's commercial banks can learn from their successful experiences and set up a management mechanism that is in line with the nation's development path. For example, projects can be clearly categorized according to the degree of risk that is defined by the Equator Principles. Moreover, applicants must be required to provide social and environmental assessment reports. At the same time, it is necessary to determine appropriate measures to deal with environmental or social violations. Enterprises must be required to regularly disclose public information. In addition, professional and technical talents within the banks should be properly trained by inviting government experts and institutional scholars as knowledge consultants, or hiring professional third-party evaluation institutions to comprehensively investigate the soundness and practicality of green projects to increase the success rate of loan projects and improve bank profitability.

Second, the green credit disclosure mechanism was established relatively late for Chinese commercial banks and aspects, such as laws and regulations, financial policies, and information communication, remain inadequate. In this case, it is difficult to generate profits in the short term and, consequently, the government must provide stronger support to green credit businesses. To this end, government departments must adjust the laws and regulations on green credit businesses and establish an independent accountability mechanism to ensure the objectivity and rationality of the implementation. In addition, financial policies should be used to strengthen industrial guidance and foster the relationship between green industry and green finance. More specifically, green credit businesses should be widely promoted through fiscal policies, the tax intensity of "three high" enterprises should be increased, and subsidies and preferential tax policies for energy conservation and environmental protection enterprises and high-tech enterprises should be strengthened. On the other hand, policies should be implemented to reasonably manage the financing interest rate of green credit projects and maintain a flexible floating range, so that the demand and supply side of funds have better market access. In addition, the government should disclose green information in a timely manner to ensure smooth communication channels among the various parties, including regulatory departments, environmental protection departments, and commercial banks. These steps could help to improve resource allocation and assist banks in improving the progress of loan projects.

Finally, China's large-scale infrastructure projects (e.g., water desalination, industrial water, sewage plant, and clean air transformation) are generally subject to long-term construction. As a result, the implementation costs for green credit policy is greater than the economic benefit in the short run. A bank can only reap the expected benefits after a project is complete and implemented. Green credit businesses can involve far more products than the current mainstream projects in China. Therefore, banks should focus on product innovation, such that green businesses can penetrate more sectors in China and, thus, effectively expand the business scope of green credit. At the same time, banks should begin by standardizing and improving the convenience of business processes, formulating clear operational guidelines and project standards, and achieving fairness and equality in project approval. This way, real energy-saving and environment-friendly enterprises could smoothly obtain financial support from banks.

**Author Contributions:** This paper was written by X.S., X.D. and R.W.; X.S. conceived the idea and designed the structure of the paper; X.S. and R.W. collected and analyzed the data and provided the technical details; X.D. wrote the draft of Section 1, Section 2, and Section 5, while X.S. and R.W. wrote Sections 3 and 4; X.D. made a final revision of the entire paper.

**Funding:** This research was funded by Beijing Language and Culture University scientific research project (Central university special funding for basic scientific research business) "Empirical research on industrial structure adjustment effect of China's carbon emissions trading market" (No. 18yj040001).

**Acknowledgments:** We would like to thank Editage (www.editage.cn) for English language editing.

**Conflicts of Interest:** The authors declare no conflict of interest.

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
