# Peer review of "Comparing the Influence of Green Credit on Commercial Bank Profitability in China and Abroad: Empirical Test Based on a Dynamic Panel System Using GMM"

_ijfs, doi:10.3390/ijfs7040064_

Round 1

Reviewer 1 Report

Line 100-111: The authors claim that the literature found a negative connection between green lending and banks’ profitability. This is not correct. There is a number of studies that found positive connections. See, for instance:

Cui, Y., Geobey, S., Weber, O., & Lin, H. (2018). The Impact of Green Lending on Credit Risk in China. Sustainability, 10(6), 2008. doi:doi.org/10.3390/su10062008

Weber, O. (2017). Corporate sustainability and financial performance of Chinese banks. Sustainability Accounting, Management and Policy Journal, 8(3), 358-385. doi:10.1108/SAMPJ-09-2016-0066

In general, the literature review could be a bit more detailed. There is a lot of research that exists in the field that is not mentioned.

Figure 2 and the paragraph describing the figure is not related to the rest of Chapter 3. What is the connection? Please explain.

Line 209: The authors claim: Since the development level and green credit business practices of China’s commercial banks 210 are inconsistent with those of international banks, we use two dynamic panel data models.

Why are they inconsistent? Please explain. Furthermore, why dynamic panel data models? What is the connection to inconsistency?

Data sources: Why are Equator Principles banks are used as comparison? The Equator Principles are a standard for project finance and not for lending. These are two very different products. Hence, credit risks cannot be compared to the three risk classes that are used by the Equator Principles.

Model building: Why do the authors finally use ROA as their variable? What is the rationale? The heading of the paper mentions profitability.

The negative impact of green lending on ROA might be a result of inter-correlation between the independent variables. What is the results for univariate regression? Univariate regression for all variables should be presented.

The main results of the paper seems to be that past ROA influences current ROA. This is not really new and interesting.

The conclusions are not related to the literature. What do other studies say? Is the current result in contrast or in-line with other studies? I am not convinced that the negative correlation between green lending and ROA is a real effect. There are many studies suggesting the opposite result.

Author Response

Response to Reviewer 1 Comments

Point 1:

Line 100-111: The authors claim that the literature found a negative connection between green lending and banks’ profitability. This is not correct. There is a number of studies that found positive connections. In general, the literature review could be a bit more detailed. There is a lot of research that exists in the field that is not mentioned.

Response 1:

Based on the reviewer’s recommendations, we have updated the references to include those with both positive and negative conclusions. You can find the updated list in the reference section

Point 2: Figure 2 and the paragraph describing the figure is not related to the rest of Chapter 3. What is the connection? Please explain.

Response 2:

Figure 2 is meant to explain hypothesis 3.

“Hypothesis 3. The growth of green credit businesses improves bank social reputation, and therefore, enhances their profitability. According to marginal cost and revenue analysis theory, a commercial bank’s marginal cost (MC) and marginal revenue (MR) curves generally intersect at point A, at which the bank’s profit is maximized, and P and Q are the equilibrium price and quantity respectively. Growth in the size of green credit generates positive externalities on related banking products, credit risk, social image, and comprehensive competitiveness. In this case, the marginal revenue curve will move up to MR’ and intersect MC at A’, causing the equilibrium price and quantity to rise to P’ and Q’ respectively. In sum, the overall profitability of the bank increases.”

Point 3: Line 209: The authors claim: Since the development level and green credit business practices of China’s commercial banks are inconsistent with those of international banks, we use two dynamic panel data models. Why are they inconsistent? Please explain. Furthermore, why dynamic panel data models? What is the connection to inconsistency.

Response 3:

We have added the following text to the introduction:

“Indeed, China has successively introduced strategies and policies related to green credit in the past decade, aiming to build a preliminary framework of using financial leverage to regulate the demand for energy conservation and emission reduction in the environment industry. However, compared to the international norms of corporate social responsibility (CSR) and industry standards of green credit, China’s framework still seems to be incomplete, and there are many differences in the rules of specific green project approval. Because of these complexities, research on the impact of the development of green credit business on commercial banks in China has not yet come to a clear consensus.”

 We hope to compare China’s green credit with other countries’ loan projects under the existing equatorial principles so as to draw some preliminary conclusions for further study in the future.

We have also added the following text to part 4:

“According to the theoretical analysis presented thus far, green credit affects commercial bank profitability through multiple channels, mostly by using time serials data or panel data. However, economic theory tells us that the bank profit is a continuous dynamic process, so the profitability from the previous period would have some impact on the current period; thus, we introduced a lag dependent variable that conforms more accurately to the theory and reality. However, once the lag dependent variable is introduced into the equation, the original static model is transformed into a dynamic model. It is only by using the dynamic panel data model that the accuracy of the conclusion can be guaranteed. Therefore, this paper established two models: one for the Chinese banking industry and one for banks from other countries. To measure the influence of green credit more effectively, we adopted a dynamic panel data model based on generalized moment estimation by introducing the lag term of ROA.”

Point 4: Data sources: Why are Equator Principles banks are used as comparison? The Equator Principles are a standard for project finance and not for lending. These are two very different products. Hence, credit risks cannot be compared to the three risk classes that are used by the Equator Principles.

Response 4:

The Equator Principles were designed to provide financial standards to assess, measure, and track the environmental risks of credit loans, thereby assisting in solving the problems of instability and irregularity in environmental project financing for financial institutions. To date, nearly 90 financial institutions around the world have formally adopted the code, including just two commercial banks from China.

However, there are only 22 listed commercial banks in China, and we could only select 12 of them that publicly disclose the amount of green credit loans for our sample. By contrast, international banks seldom announce a specific amount for equator loans; instead, they typically only disclose the aggregate project amount for three risk levels: level A (high risk), level B (medium risk), and level C (low risk).

We admit that there are differences between green credit and equator loans in terms of the contents of indicators. However, we established two separate panel data models that are treated differently rather than mixing them together. Moreover, the results of our comparative analysis shows a gap between China’s green credit business and that at the international level; this can be used to better balance the Chinese banks’ profit objectives and corporate social responsibilities.

In addition, our research object focuses on profitability rather than risk. Thus, the difference of classification standards between the Equator Principles and green credit’s risk has little impact on our research.

Point 5: Model building: Why do the authors finally use ROA as their variable? What is the rationale? The heading of the paper mentions profitability.

Response 5:

Yes, there are numerous indicators that reflect the profitability of commercial banks such as return on total assets (ROA), return on equity (ROE), and operating profit margin.

Commercial banks earn most of their income from interest rate spreads between deposits and loans in addition to highly leveraged business models. Among other factors, such as foreign exchange services, intermediate services, and security investment, this study uses ROA as the discussed variable to measure bank profitability. In addition, there are several pieces of cited research (Yang 2008; Wang 2011) that also used ROA to represent banks’ profitability.

Point 6: The negative impact of green lending on ROA might be a result of inter-correlation between the independent variables. What is the results for univariate regression? Univariate regression for all variables should be presented. The main results of the paper seems to be that past ROA influences current ROA. This is not really new and interesting. The conclusions are not related to the literature. What do other studies say? Is the current result in contrast or in-line with other studies? I am not convinced that the negative correlation between green lending and ROA is a real effect. There are many studies suggesting the opposite result.

Reference 6:

We believe that univariate regression is not necessary in our paper because the results from such an analysis were not ideal. It might be that our empirical results are not particularly refreshing to the reviewers, but the paper validates the conclusions of the existing literature to some extent while also extending the age of the data. In actuality, the conclusions from other studies are not uniform, as their data come from different countries, and, consequently, the specific results rely on different datasets.

Reviewer 2 Report

Introduction should be extended. Please clarify how your paper contributes to the existing literature and which is the motivation of your study.

The theoretical background is well presented but it lacks of literature references. For instance, you can have a look at:

Brogi, M., & Lagasio, V. (2019). Environmental, social, and governance and company profitability: Are financial intermediaries different?. Corporate Social Responsibility and Environmental Management, 26(3), 576-587.

Malik, M. S., Ali, H., & Ishfaq, A. (2015). Corporate social responsibility and organizational performance: Empirical evidence from banking sector. Pakistan Journal of Commerce and Social Sciences, 9(1), 241–247.

Maqbool, S., & Zameer, M. N. (2018). Corporate social responsibility and financial performance: An empirical analysis of Indian banks. Future Business Journal, 4(1), 84–93. https://doi.org/10.1016/j. fbj.2017.12.002

The analysis is interesting and adequate, but the sample of investigation is very small. You should include some more banks or at least provide an adequate and strong motivation supporting your choice in selecting such a small sample.

The conclusion section supports your findings, but you should also include shortcomings and limitations of your analysis providing a motivation for possible future researches inspired by your analysis.

Author Response

Response to Reviewer 2 Comments

Point 1: Introduction should be extended. Please clarify how your paper contributes to the existing literature and which is the motivation of your study. The theoretical background is well presented but it lacks of literature references.

Response 1:

We have updated the references based on the reviewer’s recommendations. You can see the updates in the bibliography.

Point 2: The analysis is interesting and adequate, but the sample of investigation is very small. You should include some more banks or at least provide an adequate and strong motivation supporting your choice in selecting such a small sample.

Response 2:

We focus on large-scale listed commercial banks; there are a limited number of such banks in China that have available data. We additionally looked for overseas commercial banks with similar scales of assets that adhere to the Equator Principles and have openly announced their data.

       We explained this challenge with the data sample in the limitations, and have revised the paper to clarify this point and provide an explanation for why our data sample is limited.

“First, the green credit business in China developed relatively late compared to that in Western countries; China’s banking industry did not begin to strengthen the disclosure of green credit data until 2008. Thus, this study examines data from the first quarter of 2008 to the fourth quarter of 2015. Second, there are only 22 listed commercial banks in China, and we were only able to select 12 banks that publicly disclose green credit information. Additionally, we selected seven equator banks in other countries that publicly disclose loan sizes in line with the Equator Principles. According to the equator principles, international banks seldom announce a specific amount for equator loans; instead, they only disclose the aggregate project amount for three risk levels: level A (high risk), level B (medium risk), and level C (low risk).”

Point 3: The conclusion section supports your findings, but you should also include shortcomings and limitations of your analysis providing a motivation for possible future researches inspired by your analysis.

Response 3:

We have updated the final sections of the paper; you can see our revisions in the conclusion section.

       “The main limitation of this paper derives from its sample size. In particular, international banks seldom announce a specific amount for their equator loans; instead, they only tend to disclose the aggregate project amount for three risk levels: level A (high risk), level B (medium risk), and level C (low risk). In the future, with a higher degree of data disclosure, subsequent scholars will be able to collect data from more international commercial banks with a similar scale of assets. Additionally, we would be able to select a more comparable variable regarding green credit or CSR and analyze its effects multi-dimensionally, including the bank's profitability, risk, and liquidity.”

Round 2

Reviewer 2 Report

Dear Authors 

thanks for improving the paper as per the requests. I am now totally fine with this version of the manuscript and I think it is ready for publication. 

Author Response

With a sharp increase in implementation of CSR practices worldwide, researchers have studied the relationship between CSR and the financial performance of financial intermediates. Numerous studies suggest that the adoption of Equator Principles or the development of green credit can improve banks’ profitability. Such as, Eisenbach et al. (2014) study the performance of large-scale financial institutions of various countries in the global project financing market and show that financial institutions adopting the Equator Principles have positive excess returns in terms of market share and other aspects. Li and Wang (2014) analyze 10 equatorial banks and find that the amount of financing by financial institutions in low-risk projects, infrastructure, renewable energy, and emerging markets is positively correlated with profitability. Malik and colleagues (2015), Brogi and Lagasio (2019), Brogi and Lagasio (2019), Maqbool and Zameer (2018) found a positive relationship between awareness of CSR activities and Pakistan, US and Indian banks’ financial performance. In terms of China’s cases, Sun and colleagues (2017) and Liao and colleagues (2019) empirically found that green credit positively affects the operational efficiency of banks in the short term. However, some other studies have found that green credit has an uncertain or even negative effect on bank profitability. Researchers including Scholtens and Dam (2007) and Wright (2012), Hu and Zhang (2016) and Han (2017), specialized green credit only increases operating costs, resulting in a decline in operating efficiency, especially in the short term. From an empirical perspective, Wang and Zhu (2017), Zhang and Dong (2018) stated that when the green loan’s proportion is high, the crowding-out and risk amplification effects gradually appear and bank performance declines. In summary, there is a general consensus in the literature that, the influence of green credit on profitability of the banking industry will become increasingly significant in the future. However, findings on the influence of adopting the Equator Principles or of promoting green credit business on bank profitability remain inconsistent, especially in the short term. In actuality, the conclusions from current studies are not uniform, as their data come from different countries, and, consequently, the specific results rely on different datasets. And our paper’s result is in line with Wang and Zhu (2017), Zhang and Dong (2018) and other related researches.
